# The Enigma of the Muon and Tau Solved by Emergent Quantum Mechanics?

**Theo van Holten**

Aerospace Engineering, Delft University of Technology, Utrechtseweg 43, 1213 TL Hilversum, The Netherlands; holtstolk.28@gmail.com; Tel.: +31-(0)357721155

**Abstract:** This paper addresses the long-standing question of how it may be explained that the three charged leptons (the electron, muon and tau particle) have different masses, despite their conformity in other respects. In the field of Emergent Quantum Mechanics non-singular electron models are being revisited, and from this exploration has come a possible answer. In this paper a deformable droplet model is considered. It is shown how the model can be made self-consistent, whilst obeying the laws of momentum and energy conservation as well as Larmor's radiation law. The droplet appears to have three different static equilibrium configurations, each with a different mass. Tentatively, these three equilibrium masses were assumed to correspond with the measured masses of the charged leptons. The droplet model was tuned accordingly, and was thereby completely quantified. The dynamics of the droplet then showed a "De Broglie-like" relation $p = K/\lambda$. Beat patterns in the vibrations of the droplet play the role of the matter waves of usual quantum mechanics. The value of $K$, calculated by the droplet theory, practically equals Planck's constant: $K \cong h$. This fact seems to confirm the correctness of identifying the three types of charged leptons with the equilibria of a droplet of charge.

**Keywords:** emergent quantum mechanics; electron; muon; tau particle; droplet model; De Broglie formula; Planck's constant

## 1. Introduction

The problem is well known: why do three types of charged leptons exist (the electron, the muon and the tau particle), and what determines the mass ratios between them? An anecdote about the famous physicist Rabi tells that, after the discovery of the muon by Anderson (1936), his reaction was "who ordered that?". Later, in 1964 Feynman [1] in his "Lectures on Physics" remarks: "Therefore, whenever someone finally gets the explanation of the mass of an electron, he will then have the puzzle of where a muon gets its mass. Why? Because whatever the electron does, the muon does the same—so the mass ought to come out the same." Still later, in the 1986 book by Abraham Pais [2] "Inward Bound; of matter and forces in the physical world" a chapter is included with the telling title "Divine laughter: the muon". The matter has been mentioned often in the literature, but remains up to this day an unsolved problem.

By a bit of serendipity a possible answer was recently found in the field of Emergent Quantum Mechanics (abbreviated Em.QM hereafter). In Em.QM an attempt is made to find a connection between quantum mechanics (QM), on the one hand, and classical mechanics, on the other, down to scales that are presently considered to be purely the realm of QM. The relation between Em.QM and QM is analogous to the relation between kinetic gas theory and thermodynamics, the latter is excellent for actual calculations, whilst the former reveals the connection with classical mechanics. In the case of Em.QM, it is hoped that in this way interpretations of quantum phenomena can be given that are less "magical" than the usual descriptions and—more importantly—that causality can be brought back to the atomic world. Most people know about Em.QM through the experiments by Couder et al. (with an early, widespread publication in 2005 [3]) where droplets bouncing on the surface of a vibrating tray of

oil exhibit double-slit diffraction, quantised levels and tunnelling. In addition, pictures and movies about these effects by other researchers like Moláček and Bush [4] are well known. These experiments show that the classical laws of nature can conspire in such a way that quantum-like behaviour is displayed. A theoretical explanation by Brady and Anderson [5] shows how it is possible that these macroscopic oil droplets, obeying the classical laws, nevertheless are good analogues of quantum particles. The same authors in another publication [6] demonstrate that a classical, macroscopic system can even display non-local behaviour similar to entangled quantum systems. A recent publication by Van Holten [7] shows that not only analogues of quantum behaviour may be obtained by Em.QM; even full replicas (quantitatively correct!) of electron behaviour inside potential wells can be found just using the classical Maxwell and Newton laws applied to a suitable model of the electron.

The electron model that has been developed in [7] might be viewed as an offspring from Lorentz's model of electrons [8]. The essential characteristic of the models in Lorentz's time was that charge was distributed within a small volume, in contrast to the singular model of today. In particular, the model Lorentz himself studied consisted of a spherical shell of charges, deposited on an insulator core, which provided containment. The model resulted in a few striking successes, but also suffered some fundamental problems. One of the successes was the analysis of so-called radiation resistance (see Section 8). On the other hand, there were problems with non-compliance with special relativity, and anti-causal dynamic behaviour like pre-acceleration and runaway motion. Poincaré [9] discussed how Lorentz's theory could be made to agree with special relativity. While non-singular models were abandoned, the interest in them continued (as related by Feynman [1]), if only out of curiosity about what the cause of their fundamental problems was. Remarkably, all the remaining problems were relatively recently (1992) cleared up and solved by Yaghjian [10].

The model presented below has its origin in a study of the dynamic behaviour of a charged macroscopic cloud. Triggered by several intriguing perturbation terms in the equations of motion, the model was subsequently scaled down to the size of electrons. The model of electrons thus obtained may, despite the absence of a separate core, be considered to be a deformable version of the relativistically rigid charge distributions studied by Lorentz. Thanks to this fact, the analysis of the model owes much to Lorentz and Yaghjian's theories. Alternatively, the model may also be viewed as an electron subject to "zitter", after having simplified the model by time-averaging (Figure 1). As discussed in Section 2, just time-averaging would be an incomplete and actually wrong way to obtain a simplified model, since it artificially introduces an explosion tendency which is not real. Complementary to the time-averaging, an apparent force should be included to obtain a consistent simplification of the real situation. All this leads to a droplet model, with a surface-tension-like containment force (not a real physical force but an apparent force belonging to the level of the time-averaged model only). The basic modeling principles are discussed in Section 2, and further model assumptions are described in Section 3. An unexpected by-product of this model was that the deformable droplet of charge appeared to have three different equilibrium configurations (only one of them stable), each associated with a different value of the mass. A discussion of the equilibrium conditions is given in Section 12.

Tentatively, the three equilibrium configurations were assumed to correspond to the three types of charged leptons, and the model was tuned accordingly (Section 13). Having thus fully quantified the model, other properties of the model, not related to the static equilibria, were sought that quantitatively could be verified. It appeared that the dynamic behaviour of the model is in agreement with De Broglie's relation $p = h/\lambda$, whereas the quantitative analysis by Em.QM provides the correct numerical value of Planck's constant in this formula. The assumption that the electron, muon and tau correspond to the three equilibria of a droplet thus seems to be justified (strictly within the context of Em.QM).

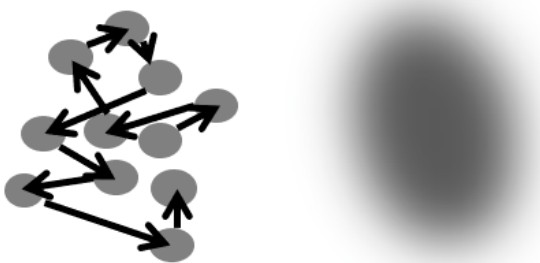

**Figure 1.** The "zitter" of an electron (left) and a time-averaged view of it (on the right). Time-averaging is an incomplete model simplification: It introduces an unrealistic explosion tendency.

The rigorous mathematical derivation of the electromagnetic self-forces is too long for the present article, and interested readers are referred to the mathematical appendix of [7]. However, the self-forces are heuristically discussed, and their form and magnitude are made plausible by invoking the laws of energy and momentum conservation, as well as the known radiation laws. A summary of the model's dynamics may be found in Sections 5 and 10 in the form of a set of equations of motion for the deformable droplet. The subsequent derivation of De Broglie's relation and Planck's constant is finally shown in Section 16.

## 2. Principles of the Model

Inherent in any model consisting of a distributed charge is the problem of how an explosion due to the mutual repulsion forces between the charge elements is prohibited. The presently-investigated model is assumed to be obtained by a time-averaged observation of an electron subject to the "zitter" motion. This basic idea is sketched in Figure 1. "Zitter" (a German term introduced by Schrödinger in 1930, meaning "shudder" or "tremble") is the phenomenon that an electron continually performs tiny jumps in all directions. A time-averaged picture of the "zittering" electron is then something like sketched schematically on the right in Figure 1. Perhaps it could also be compared with what is known as a "dressed" or "clothed" electron. It should be realised that analysing the resulting cloud of charge by classical electromagnetic theory shows up an explosion tendency, which is not present in the original "zittering" electron on the left in Figure 1. In the case of the real, zittering electron, at any instant of time there is no more than one charged particle present, so that there is no tendency of the "zitter volume" to explode. At most this volume may slowly drift away. On the other hand, the time-averaged model shows the tendency to explode. The explosion tendency is clearly not real, and is just artificially introduced by the time-averaging. In order to obtain a consistent model that reflects reality, and that allows an unrestricted application of the electromagnetic laws, the addition of apparent forces to complete the model simplification is necessary. Feynman [1] uses the terminology "Poincaré force" to indicate any kind of containment force "taming" the explosion tendency of a smeared out charge. In our case the Poincaré force is not a real physical force, it is just an apparent force that forms an integral part of the modelling process, and occurs only on the level of the time-averaged model world. The form such apparent forces in our case should take must be some kind of surface tension, since the explosion tendency is largest in the outer shells of the charged cloud, where the repulsion all comes from within the cloud. The same conclusion is also reached [7] when the binding energy is considered by a procedure similar to the one adopted by Von Weiszäcker concerning the droplet model of atomic nuclei (see a summary of this approach in, e.g., the textbook by Alonso and Finn [11]). It should be stressed that just the procedure of Von Weiszäcker is copied; the surface-tension-like force has nothing to do with the strong nuclear force. The apparent force is not even physically real, it is no more than a modelling tool.

A model consisting of a "smeared out" charge distribution completed by a surface-tension-like apparent force will be called a droplet model of the electron.

## 3. Model Assumptions and Notations

The droplet model served as a working hypothesis to explore what would be the consequences of assuming a non-singular electron model. For this reason, drastic simplifying assumptions were introduced in order to facilitate an easy analysis with the limited aim to just get a first indication of the consequences. The model analysed is shown in Figure 2. The droplet is assumed to move along the Z-axis inside a one-dimensional potential well, one half of which is schematically indicated by the barrier at a distance $a/2$ from the well centre at $z = 0$. What is sketched suggests a potential box, but the barrier may be soft as well, such as in a parabolic well. The motion of the droplet, as well as the motion of all its elements, is likewise assumed to be one-dimensional. Consequently, the charge elements have a velocity in the Z-direction only. The charge distribution is assumed to have rotational symmetry around the Z-axis, and fore-aft symmetry w.r.t. an "equator plane". We can thus define a midpoint which is the centre of mass as well as the centre of charge.

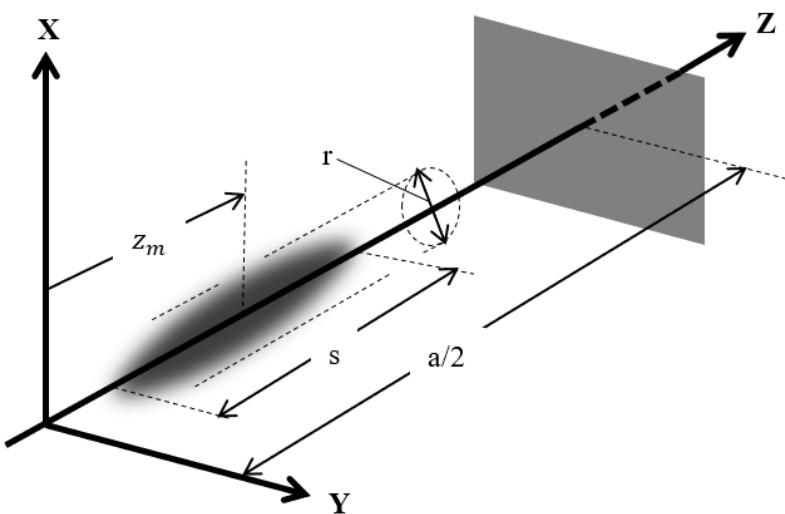

**Figure 2.** Configuration and notations.

The model has two degrees of freedom: The variable position of the midpoint $z_m(t)$, and the elongation $s(t)$. A quasi-static deformation mode is assumed, where stretching (i.e., an increase of the total length $s$) takes place as in a spring without dynamic effects. For each charge element the relative position w.r.t. the midpoint, expressed as a fraction of the total length, is therefore constant:

$$\frac{z_{elem}(t) - z_m(t)}{s(t)/2} = constant \tag{1}$$

The constant in this expression and the constant values $x_{elem}$ and $y_{elem}$ together form a set of "labels" which identify a particular element of the total charge, no matter the changes in time of $z_m(t)$ and $s(t)$. The velocity, acceleration and jerk of any element can, by Equation (1), be determined in terms of $\dot{z}_m$, $\ddot{z}_m$, ... and $\dot{s}$, $\ddot{s}$, ... by differentiation.

A final assumption is that the droplet has an elongated shape in the Z-direction (i.e., varying from a Zeppelin-shape to a needle or even a long wire or string), so that the analysis of the electromagnetic field and the self-forces can make use of "slender-body" approximations.

### 4. Matched Asymptotic Expansion Procedure to Determine the Electromagnetic Field

Determining the electromagnetic field and the electromagnetic self-forces on the droplet requires a lengthy analysis, too long for this article. The interested reader is referred to the appendix of [7] where the complete derivations are given. In the present article the final results will be shown in the form of the equations of motion, with a discussion of all the terms. The analysis in [7] uses as a mathematical tool a so-called "matched asymptotic expansion method for singular perturbation problems", an approach that was popular in fluid dynamics in the 1960s and 1970s, but was later forgotten when digital computation took over. For the near field as well as the far field, separate asymptotic expansions are determined, with $(a/\lambda)$ as expansion parameter, where $a$ is the width of the potential well, and $\lambda$ is a typical wavelength in the electromagnetic field. An equivalent expansion parameter is $(v/c)$. Through the mediation of a "common field", the expansions of the near and far field are matched with each other. The expansions are continued up to and including terms of the order $(v/c)^3$. This accuracy is chosen so that the radiation resistance (Section 8) acting on the droplet is in agreement with the power radiated to infinity. It will be seen below that the well dimension $a$ does not occur in the expressions for the self-forces. Therefore, up to the order $(a/\lambda)^3$ (or: $(v/c)^3$) the results are applicable to freely moving charges as well.

The analysis is non-relativistic, as far as possible, in view of the use of Maxwell's equations that inherently include relativistic effects. The choice of which terms to include depends strictly on the asymptotic order, and does not depend on whether or not the term indicates a relativistic effect.

### 5. Equation of Motion for the Translation

The resulting equation of motion for the translation in the Z-direction is given below, followed by a discussion of the terms occurring in this equation:

$$\left[ m_{bare} + 2 \, \frac{U_{es}(s)}{c^2} \right] \dot{v}_m - \frac{2}{3} \frac{q^2}{4\pi\varepsilon_0} \frac{\ddot{v}_m}{c^3} \; = \; m_{em}(s) \, . v_m \frac{\dot{s}}{s} + q . E_{external} \tag{2}$$

The coefficient of the acceleration $\dot{v}_m$, i.e., the effective mass of the droplet, consists of two separate contributions. The first term $m_{bare}$ between the square brackets in Equation (2) denotes what Lorentz called the "material mass" of the droplet. We follow in the next sections Yaghjian's discussion of $m_{bare}$ [10], in which he introduced the nomenclature "bare" (meaning "weighty") for the material mass that is intrinsically bound up with charge (the index "bare" should not be confused with what is often called a "bare (i.e., *un*dressed) electron").

### 6. Electromagnetic Mass

The second term of the effective mass depends on the so-called electrostatic energy $U_{es}$. $U_{es}$ denotes the formation energy of the droplet, i.e., the energy that must be expended if all the charge elements were brought together from a large distance, against the mutual repulsion forces. According to electrostatic theory [1]:

$$U_{es} = \frac{1}{2} \iiint_{droplet} \sigma . \; \Phi_{el.static} \; dx.dy.dz \tag{3}$$

where $\sigma$ denotes the charge density, and $\Phi_{el.static}$ the electrostatic potential. The evaluation of $U_{es}$ shows for the droplet [7]:

$$U_{es}(s) = \frac{1}{2} \frac{q^2}{4\pi\varepsilon_0} \frac{1}{s} \alpha \tag{4}$$

where $q$ is the total charge of the droplet, and the non-dimensional factor $\alpha$ is a constant of the order $O(1)$ depending on how the charge density varies throughout the droplet. In the present theory, no assumptions have been made as to the distribution of the charge density.

The part of the inertial mass that is associated with electrical charges was in Lorentz's time baptised the "electromagnetic mass" $m_{em}$. In the case of the droplet model, its value, in view of Equations (2) and (4), is:

$$m_{em}(s) = 2 \frac{U_{es}(s)}{c^2} = \frac{1}{c^2} \frac{q^2}{4\pi\varepsilon_0} \frac{1}{s} \alpha \qquad (droplet\ model) \tag{5}$$

In Lorentz's electron theory, a similar inertial term occurs, where the fixed radius of the sphere replaces the variable length $s$ in Equation (5), and $\alpha = \frac{2}{3}$ (*Lorentz model*). For his relativistically rigid, spherical shell model of electrons, he found $m_{em} = \frac{4}{3} \frac{U_{es}}{c^2}$, which at the time caused some concern because it clearly does not conform to the relativistic relation between mass and energy. For other distributions of the charge, the factor $\alpha$ will be different, but, because of its role as non-dimensional "form-factor", $\alpha$ is in general expected to have a value of the order $O(1)$.

## 7. Poincaré's Binding Energy

Yaghjian [10] requires that the above expressions are in agreement with $E = mc^2$, when $m$ is taken as the total mass $m_{bare} + 2 \frac{U_{es}(s_{equ})}{c^2}$, and if $E$ equals the electrostatic energy (the energy of formation) $U_{es}(s_{equ})$. The condition that $s$ must be taken equal to the static equilibrium length, i.e., $s = s_{equ}$, has been added by the present author because Yaghjian does not consider deformable configurations. By "static equilibrium length" $s = s_{equ}$ is meant that the velocity $v_m = 0$ and that there are no external squeezing forces. Under these conditions the electrostatic explosion tendency is balanced by the squeezing due to the surface tension. Yaghjian's conclusion is that one can satisfy the relativity requirement if $m_{bare} < 0$:

$$m_{bare} = -\frac{U_{es}(s_{equ})}{c^2} = -\frac{1}{2} m_{em}(s_{equ}) \tag{6}$$

The total effective mass $m_{bare} + 2 \frac{U_{es}(s_{equ})}{c^2}$ is always positive, but $m_{bare}$ indicates a mass deficit in this positive sum. Such a mass deficit must be associated with the binding energy that is present in a charge distribution with cohesion, as opposed to a loose collection of charge elements kept together by a distribution of forces. The interpretation based on binding energy is in agreement with Poincaré's analysis [9] of Lorentz's model, where Poincaré came to the same conclusion, viz. that the spherical shell model of Lorentz can only be in agreement with relativity theory when the binding forces between the shell of charges and the core of insulator material are taken into account. The effective mass in Equation (2) may thus alternatively be written:

$$m_{bare} + 2 \frac{U_{es}(s)}{c^2} = -\frac{1}{2} m_{em}(s_{equ}) + m_{em}(s) \tag{7}$$

Whereas the mass of the droplet under static equilibrium conditions equals:

$$m_{equ} = \frac{1}{2} m_{em}(s_{equ}) \tag{8}$$

## 8. Agreement with Larmor's Radiation Law

Returning to the equation of motion, Equation (2), the second term $-\frac{2}{3} \frac{q^2}{4\pi\varepsilon_0} \frac{\dddot{v}_m}{c^3}$ derived in [7] is exactly in agreement with the results found by Lorentz. The term was called "radiation resistance" (sometimes "radiation reaction"), and may be viewed as a recoil force in reaction to radiation. The radiation energy escaping to infinity is according to Larmor's law:

$$\frac{dW}{dt} = \frac{2}{3} \frac{1}{c^3} \frac{q^2}{4\pi\varepsilon_0} (\dot{v})^2 \tag{9}$$

This is in agreement with the work done by an external force that overcomes the radiation resistance if sinusoidal motion is assumed and when one averages over a cycle. In the matched asymptotic analysis of the self-forces [7], this recoil force is only found when terms are included of at least the order $O(v/c)^3$, which explains the choice to continue the expansions up to and including the third order of the expansion parameter.

## 9. Conservation of Momentum

In the right-hand side of the equation of motion, Equation (2), another self force $m_{em}(s) . v_m \frac{\dot{s}}{s}$ is seen, which couples the translation motion and the variable elongation $s$. The term arises from the rigorous mathematical analysis [7], but can easily be understood since, thanks to this term, the equation of motion can be written in the form:

$$\frac{dp}{dt} = \frac{2}{3} \frac{q^2}{4\pi\varepsilon_0} \frac{\ddot{v}_m}{c^3} + q.E_{external} \tag{10}$$

where $p$ denotes the momentum of the droplet, based on its instantaneous effective mass:

$$p = \left[ -\frac{1}{2} m_{em}(s_{equ}) + m_{em}(s) \right].v_m \tag{11}$$

It is thus seen that the equation of motion is in agreement with Newton's momentum equation (non-relativistic, due to the cutting off of terms of order $O(v/c)^4$ and higher in the asymptotic expansions).

## 10. Equation of Motion for the Pulsation

The length variations $s(t)$ will be called the "pulsation motion" of the droplet. The equation of motion for this degree of freedom is:

$$\left[ m^*_{bare} + 2 \frac{U^*_{es}(s)}{c^2} \right] \ddot{s} = - \left[ 1 - \frac{v_m^2}{c^2} \right].\frac{\partial U_{es}}{\partial s} + Q_{s,surf.tension} + Q_{s,external} \tag{12}$$

which will now be discussed term by term.

### 10.1. Generalised Electrostatic Energy and Generalised Squeezing Inertia

The self-forces in this equation are generalised forces in the s-direction, defined as in Lagrange's formulation of dynamics. This entails that the forces on the charge elements should be multiplied by the weight $\left(\frac{z-z_m}{s}\right)$ when determining the generalised self-forces $Q_s$. The electrostatic energy $U^*_{es}(s)$ in the equation of motion is similarly defined as the formation energy of a charge density $\sigma. \left(\frac{z-z_m}{s}\right)$, where $\sigma$ is the actual charge density in the droplet (see for comparison Equations (3) and (4)):

$$U^*_{es}(s) = \frac{1}{2} \iiint_{droplet} \sigma. \left(\frac{z-z_m}{s}\right) \Phi^*_{el.static} \, dx.dy.dz = \frac{1}{2} \frac{q^2}{4\pi\varepsilon_0} \frac{1}{s} \alpha^* \tag{13}$$

where $\Phi^*_{el.static}$ is the electric potential field of the modified charge density $\sigma. \left(\frac{z-z_m}{s}\right)$.

In general, one may expect the non-dimensional factor $\alpha^*$, defined by Equation (13), to be an order smaller than the factor $\alpha$ occurring in Equation (4), see [7]. The generalised mass in the pulsation Equation (12) may, by similar arguments as the effective mass in Equation (2), be written:

$$\left[ m^*_{bare} + 2 \frac{U^*_{es}(s)}{c^2} \right] = \left[ -\frac{1}{2} m^*_{em}(s_{equ}) + m^*_{em}(s) \right] \tag{14}$$

where:

$$m_{em}^*(s) = 2 \frac{U_{es}^*(s)}{c^2} = \frac{1}{c^2} \frac{q^2}{4\pi\varepsilon_0} \frac{1}{s} \alpha^* \tag{15}$$

*10.2. Generalised Surface-Tension Force*

The generalised force expressing the surface-tension-like force has the general form:

$$Q_{s,surf.tension} = A.\left(s - s_{sphere}\right) + C.\frac{1}{s} = A.s + B + \frac{C}{s} \tag{16}$$

The first part, $A.\left(s - s_{sphere}\right)$, reflects the property that surface tension causes a tendency to squeeze the droplet into a spherical shape, which tendency is larger the more the actual shape differs from a sphere. This first term is in the second line written as $A.s + B$. The second term represents a scale effect: The compression due to surface tension leads to a tendency to minimize the size, which tendency is relatively stronger the smaller the overall size. In the case of spherical droplets, the scale effect can be derived to be proportional to the inverse radius, which scale effect in the present case (one-dimensional variations of the elongation $s$) is represented by the factor $C/s$.

## 11. Conservation of Energy

The first term $-\left[1 - \frac{v_m^2}{c^2}\right].\frac{\partial U_{es}}{\partial s}$ in the right hand side of Equation (12) will now be discussed. In view of the physical meaning of $U_{es}$ (energy of formation), the term $-\frac{\partial U_{es}}{\partial s}$ evidently expresses the expansion tendency in the s-direction for the static droplet ($v_m = 0$). An alternative expression, in view of Equation (4), is:

$$Q_{s,el.static}\big|_{v_m=0} = -\frac{\partial U_{es}}{\partial s} = \frac{U_{es}}{s} = \frac{1}{2} \frac{q^2}{4\pi\varepsilon_0} \frac{1}{s^2} \alpha \tag{17}$$

which, in the complete equation of motion, Equation (12), is multiplied by the typically relativistic factor $\left[1 - \frac{v_m^2}{c^2}\right]$. This is one of the few relativistic effects that should be kept in the equation of motion, because it has an asymptotic order $O(v/c)^2$. The complete term $Q_{s,el.static} = -\left[1 - \frac{v_m^2}{c^2}\right].\frac{\partial U_{es}}{\partial s}$ indicates a coupling between the translation motion and the pulsation. Its physical meaning can be understood when considering the simplified case of a droplet where an artificial external force cancels the radiation resistance. This is a situation where the loss of energy by radiation is compensated by an external energy input, and the droplet itself must have a constant value of its energy. According to Equation (10), the momentum $p$ then is constant, say $p_0$. The energy associated with the translation velocity is purely kinetic energy:

$$E_{transl} = \frac{1}{2} \frac{p_0^2}{\left[m_{bare} + 2 \frac{U_{es}(s)}{c^2}\right]} \tag{18}$$

It is seen that $E_{transl}$ fluctuates when $s$ is variable. Changes of $E_{transl}$ cannot come from outside the system, and can only happen if there is at the same time an energy transfer to or from $E_{puls}$, the energy in the pulsation. The power that is transferred to the translation mode equals the time derivative:

$$\frac{dE_{transl}}{dt} = \frac{1}{2} \frac{p_0^2}{\left[m_{bare} + 2 \frac{U_{es}(s)}{c^2}\right]^2} \frac{2}{c^2} \frac{U_{es}(s)}{s} \dot{s} = \frac{v_m^2}{c^2} \frac{U_{es}(s)}{s} \dot{s} \tag{19}$$

The equation of motion for the pulsation, Equation (12), can be written as a power equation through multiplication by $\dot{s}$. The equation is then written in the form:

$$\left[m_{bare}^* + 2 \frac{U_{es}^*(s)}{c^2}\right]\ddot{s}\,\dot{s} = \left[Q_{s,surf.tension} - \frac{\partial U_{es}}{\partial s}\right]\dot{s} - \frac{v_m^2}{c^2} \frac{U_{es}}{s} \dot{s} \tag{20}$$

It is seen that the coupling term $-\frac{v_m{}^2}{c^2}\frac{U_{es}}{s}\dot{s}$ in Equation (20) is in agreement with the energy transferred to the translation according to Equation (19). In other words, the presence of the coupling term in the pulsation equation of motion guarantees that the total energy of the droplet is conserved.

## 12. Three Equilibrium Configurations

What will be considered now is the equilibrium between the expansion tendency by electrostatic repulsion and the squeezing by the surface-tension-like apparent forces, in a static condition of the droplet and in the absence of other forces. From the equation of motion, Equation (12), it follows, under these static conditions, substituting Equations (16) and (17):

$$\left[Q_{s,surf.tension} - \frac{\partial U_{es}}{\partial s}\right] = \left(A.s + B + \frac{C}{s}\right) + \frac{1}{2}\frac{q^2}{4\pi\varepsilon_0}\frac{1}{s^2}\alpha = 0 \tag{21}$$

which may be written as a third-order algebraic equation in $s$. If all the roots are real, there are three different solutions for the value of the elongation $s$ where equilibrium is possible. Note that $Q_{s,surf.tension}$ contains three unknown constants, $A$, $B$ and $C$, usable for tuning the model. The equilibrium conditions are shown in Figure 3, where the general shape of $Q_{s,el.static}$ and $Q_{s,surf.tension}$ (absolute values) is schematically shown as a function of the elongation $s$. Since the electromagnetic mass is inversely proportional to the elongation $s$, there are three different values of the mass associated with the three equilibrium configurations.

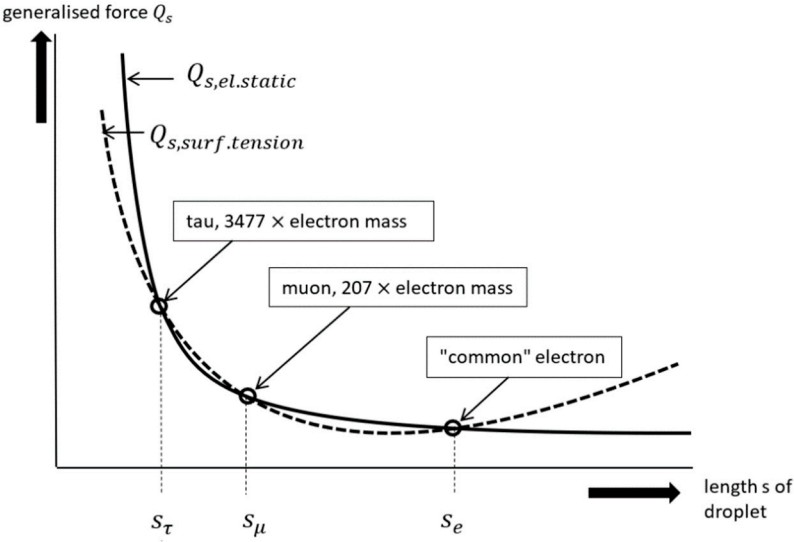

**Figure 3.** The three equilibria of the droplet, and their tentative identification with the three charged leptons.

At this point it becomes interesting to start a further investigation into whether it is allowed to identify the three equilibrium points in Figure 3 as the tau-particle, the muon and the common electron. We proceed as follows: Tentatively, we will make the assumption that the three equilibrium points indeed correspond to the three charged leptons. This fully quantifies the droplet model, since there are three undetermined constants in the equation of motion, Equation (12), as may be seen in Equation (16). We will then take a step further than the static equilibrium situation considered until now, and compare the dynamic characteristics of the droplet model with the real dynamic behaviour of electrons. If there happens to be quantitative agreement, this will be considered as a sufficient validation of the so-quantified model, as well as a confirmation that the three charged lepton masses can be explained by the droplet model.

## 13. Quantifying the Model

If the cubic algebraic equation derived from Equation (21) has three real solutions, they will be denoted (as shown in Figure 3) by the symbols $s_e$ (electron), $s_\mu$ (muon) and $s_\tau$ (tau particle). By substituting $s_{equ} = s_e$, $s_\mu$ or $s_\tau$ into Equation (22) below, the corresponding values of the mass are found. Equation (22) follows from Equations (5) and (8):

$$m_{equ} = \frac{1}{2} m_{em}(s_{equ}) = \frac{1}{2} \frac{1}{c^2} \frac{q^2}{4\pi\varepsilon_0} \frac{1}{s_{equ}} \alpha \tag{22}$$

Defining two mass ratios $\mu$ and $\tau$ as in the following, Equation (23), application of Equation (22) shows:

$$\mu = \frac{m_{muon}}{m_{electron}} = \frac{s_e}{s_\mu} \ and \ \tau = \frac{m_{tau}}{m_{electron}} = \frac{s_e}{s_\tau} \tag{23}$$

When use is finally being made of the general properties of cubic equations, we find from Equation (21):

$$\frac{B}{A} = -s_e \left( 1 + \frac{1}{\mu} + \frac{1}{\tau} \right), \ \frac{C}{A} = s_e^2 \left( \frac{1}{\mu} + \frac{1}{\tau} + \frac{1}{\mu\tau} \right) \ and \ \frac{1}{2} \frac{q^2}{4\pi\varepsilon_0} \frac{\alpha}{A} = -s_e^3 \frac{1}{\mu\tau} \tag{24}$$

These values may be substituted into Equations (12) and (16) to find a—tentatively quantified—equation of motion whose solution, by comparison with the known behaviour of electrons, can show whether the identification in Figure 3 with the three leptons indeed makes sense.

## 14. Linearised Equations of Motion

The next few steps of the derivation are not written out completely, they are straightforward and may be checked by an interested reader (or may be found in [7]). The first step is to substitute Equation (24) into Equations (16) and (12), so that the coefficients in the equation of motion, Equation (12), are expressed in terms of the known mass ratios $\mu$ and $\tau$. Next, both the equations of motion, Equations (2) and (12), are linearised by writing $s(t) = s_e \left[ 1 + \frac{\Delta s}{s_e}(t) \right]$ where $\frac{\Delta s}{s_e}$ is assumed to be small, so that squares and higher powers of $\frac{\Delta s}{s_e}$ in series expansions may be neglected. Using convenient short notations, the equations of motion transform into the following set of equations, Equations (25)–(28):

$$\frac{dp}{dt} = \gamma \ddot{v}_m + q.E_{external} \ with \ p = m. \left( 1 - 2 \frac{\Delta s}{s_e} \right).v_m(t) \tag{25}$$

The symbol $m$ here stands for the constant equilibrium mass of the electron

$$m = \frac{1}{2} m_{em}(s_e)$$

Note that the variable $v_m(t)$ has not been linearised. The linearised pulsation equation reads:

$$\frac{\ddot{\Delta s}}{s_e} + \omega_0^2 \left( 1 - 2kv_m^2 \right) \frac{\Delta s}{s_e} = -k\omega_0^2 v_m^2 \tag{26}$$

where:

$$\omega_0^2 = \frac{m}{m^*} \frac{c^2}{s_e^2} \mu\tau \ with \ m^* = \frac{1}{2} m_{em}^*(s_e) \tag{27}$$

$$k = \frac{1}{c^2} \frac{1}{\mu\tau} \tag{28}$$

In the case of a free motion without an external potential ($E_{external} = 0$) these equations describe damped simultaneous vibrations of $\frac{\Delta s}{s_e}$ and $v_m$. The coefficient $\gamma = \frac{2}{3} \frac{q^2}{4\pi\varepsilon_0} \frac{1}{c^3}$ of the damping term (i.e., the radiation resistance) is very small (with $c^3$ in the denominator), and we can get a first insight into the character of the vibrations by neglecting the damping. Physically this means that, coupled with the ultra high frequency periodic lengthening and shortening of the droplet (i.e., the periodic pulsation), the droplet moves with small jumps around a constant average velocity: $v_m(t) = v_{av} + \Delta v(t)$, where the average velocity $v_{av}$ is constant, and $\Delta v(t)$ is periodic with the same frequency as the pulsation. The final step is to substitute the translation equation, Equation (25), into the pulsation equation, Equation (26), and to strictly adhere to the linearisations:

$$\frac{\ddot{\Delta s}}{s_e} + \omega_0{}^2 \left( 1 + 2k.v_{av}{}^2 \right) \frac{\Delta s}{s_e} = -k\omega_0{}^2 v_{av}{}^2 \tag{29}$$

Note the change of sign in the second term compared with Equation (26). The frequency of the pulsation and the velocity jumps, since $k$ is small, is:

$$\omega = \omega_0 \left( 1 + k.v_{av}{}^2 \right) \tag{30}$$

with $\omega_0$ and $k$ defined by Equations (27) and (28).

### 15. Physical Meaning of the Frequency Shift Due to Velocity

Clearly, the average velocity of the droplet causes a positive shift of the pulsation frequency. The fact that the frequency increases with speed is not in contradiction with relativity theory, because relativistic effects such as time dilation will show up in the asymptotic derivations as terms of a fourth asymptotic order. Terms of this order were neglected in the present theory. The frequency shift $\Delta\omega$ due to velocity, defined as:

$$\Delta\omega = k\omega_0.v_{av}{}^2 \tag{31}$$

does have a clear physical meaning, but there is the question of how it will manifest itself in actual practice. In [7] it is shown that this shift automatically takes control of matters in the case of a droplet moving back and forth inside a potential well, and leads to energy quantisation in agreement with the usual results of QM. However, the shortest way to get to our present goal, viz. to a quantitative verification of the equations of motion, is to use the droplet theory for a derivation of De Broglie's equation. This may be done by imagining an experiment where we let the electromagnetic field induced by a static—but pulsating—droplet ($v_{av} = 0$) interfere with that due to a moving droplet. We will find low frequency beats ("low frequency" compared with the pulsation frequency) in the combined field, with the beat frequency $\Delta\omega$ according to Equation (31).

### 16. De Broglie's Relation and Planck's Constant

In our case, where we want to compare some calculated characteristics of the model with known values, we proceed as follows. Define a "beat wavelength" $\lambda_{beat}$ as the distance travelled by the droplet in the time $T = \frac{2\pi}{\Delta\omega}$ between two consecutive cycles of the beat:

$$\lambda_{beat} = v_{av}\, T = \frac{2\pi}{k\omega_0} \frac{1}{v_{av}} \tag{32}$$

Next define an "average momentum" $p = m\, v_{av}$ ($m$ and $v_{av}$ as defined in Section 14). Equation (32) may then be written in the form:

$$p = \frac{K}{\lambda_{beat}} \tag{33}$$

$$K = \frac{2\pi}{k\omega_0} m = \frac{1}{c} \frac{q^2}{4\pi\varepsilon_0} \sqrt{\mu\tau}.\beta, \; and \; \beta = \sqrt{\pi^2 \alpha\, \alpha^*} \tag{34}$$

Equation (33) is a relation remarkably similar in form to De Broglie's equation $p = h/\lambda$. We now compare the numerical value of the constant $K$ with the value of Planck's constant $h = 6.6256 \times 10^{-34}$ *J.s.* The constant $K$ in the droplet theory depends on the factor $\beta$, so that it depends on how the charge is distributed within the droplet. The precise value of the factor $\beta$ is thus unknown: In the theory as it stands now, no assumptions were introduced (nor were they needed) about how the charge density varies throughout the droplet. The only thing known is that the factor is of order unity: $\beta = O(1)$ (see the remarks about the orders of $\alpha$ and $\alpha^*$ in Sections 6 and 10). The best estimate that presently can be made is therefore $\beta = 1$. The other constants in Equation (34) have the value $c = 2.9979 \times 10^8 m/s$, $q = 1.6021 \times 10^{-19} C$, $\varepsilon_0 = 8.8544 \times 10^{-12} N^{-1} m^{-2} C^2$, $\mu = 206.85$ and $\tau = 3477.1$. The resulting value of $K$ becomes:

$$K = 6.52 \times 10^{-34} \ J.s \ for \ \beta = 1 \tag{35}$$

The difference with Planck's constant is 1.5 %. It is an error that can be considered to be well within the expected accuracy margins, in view of the exploratory nature of the derivations (one-dimensional approach, just three terms included in the matched asymptotic expansion, linearisations, etc.).

## 17. Conclusions

It may be concluded that the relation between the momentum and the wavelength of the droplet model is quantitatively in good agreement with De Broglie's relation in QM. The quantification of the droplet model was based on the assumption that the three equilibria of the charged droplet would correspond to the electron, muon and tau (i.e., the three charged leptons of the standard model). Since the quantification of the droplet model seems to be correct (in view of the fact that Em.QM is then able to reproduce the value of Planck's constant), it is plausible that the three charged leptons indeed correspond to the equilibria of a droplet of charge. This answers the first question "why do three charged leptons exist?"

In reverse, starting from the given value of Planck's constant, the analysis by Em.QM gives one relation between the two mass ratios $m_{muon}/m_{electron}$ and $m_{tau}/m_{electron}$, Equation (34). If a second relation could be found, it would fix these ratios. A second relation has not been found (yet), although it is plausible that such a relation must exist because one expects that the mass ratios of the various guises of the electron are determined by the laws of nature and are not completely arbitrary. The second question, "what determines the mass ratios?", cannot yet be answered, although a first indication in which direction to look for an answer has been found.

**Funding:** This research received no external funding.

**Conflicts of Interest:** The authors declare no conflict of interest.

**Notations:** Symbols recurring throughout the text are here listed. Non-recurring symbols are explained immediately underneath the expressions in which they are used.

| | |
|---|---|
| $a$ | width of potential well (see Figure 2), radius of Lorentz's electron model |
| $A, B, C$ | constants in surface-tension-like force, Equation (21) |
| $c$ | velocity of light |
| $E$ | field strength, energy |
| $h$ | Planck's constant |
| $k$ | constant factor defined in Equation (28) |
| $K$ | constant in "De Broglie-like" relation of Em.QM, defined in Equation (34), $K \cong h$ |
| $m$ | effective mass $\left.\frac{1}{2} m_{em}(s)\right|_{s=s_e}$ of droplet at static equilibrium (common electron) |
| $m_{bare}$ | material mass |
| $m_{em}$ | electromagnetic mass, defined in Equation (5) |
| $m_{em}{}^*$ | generalised electromagnetic mass |
| $p$ | momentum |
| $q$ | total charge of droplet, in Equation (34) equal to unit charge |

| | |
|---|---|
| $Q_s$ | generalised expansion force |
| $s(t)$ | length of droplet, see Figure 2 |
| $s_{equ}$ | static equilibrium length without external influences |
| $U_{es}$ | electrostatic energy of droplet, defined in Equation (3) |
| $U_{es}^*$ | electrostatic energy of droplet with charge distribution $\sigma$ modified into $\sigma.(z-z_m)/s$ |
| $v_m$ | velocity of midpoint, as defined in Figure 2 |
| $v_{av}$ | average translation velocity |
| $\Delta v$ | amplitude of periodic velocity perturbation |
| $\alpha$ | non-dimensional constant depending on distribution of charge within the droplet |
| $\gamma$ | short for $\frac{2}{3}\frac{q^2}{4\pi\varepsilon_0}\frac{1}{c^3}$ |
| $\varepsilon_0$ | vacuum permittivity |
| $\lambda_{beat}$ | wavelength of beats (Section 16) |
| $\mu$ | mass ratio muon/common electron |
| $\sigma$ | charge density |
| $\tau$ | mass ratio tau particle/common electron |
| $\Phi$ | potential field |
| $\omega_0$ | "zero-speed frequency", defined in Equation (27) |

Indices:

| | |
|---|---|
| $av$ | time averaged |
| $e$ | pertaining to electron |
| $equ$ | pertaining to static equilibrium without external forces |
| $\mu$ | muon |
| $\tau$ | tau particle |

Superscripts:

| | |
|---|---|
| $*$ | pertaining to modified charge distribution $\sigma.\left(\frac{z-z_m}{s}\right)$ |

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
