# Peer review of "The Enigma of the Muon and Tau Solved by Emergent Quantum Mechanics?"

_applsci, doi:10.3390/app9071471_

Round 1

Reviewer 1 Report

Should certainly be addressed:

The ‘apparent’ force that keeps the charged droplet from ‘exploding’ should be discussed. What is the origin of this force? In contrast to what the author states, this surface tension or binding energy is not similar to the atomic nucleus droplet model, as there the strong nuclear force keeps the quarks confined. What, on the other hand, confines the Zitter movement, or over what time scales does one have to average to obtain the droplet model (i.e. does the droplet binding become apparent)? Does this time scale agree with the equations of motion of the droplet?

From this work the equilibrium droplet sizes for the charged leptons could be calculated. Why is this not done? The values should be verified with respect to e.g. the Planck length, and discussed with respect to the ‘slender-body’ approximation that is made (lines 147-149).

The author should discuss what the impact of the potential well is on his conclusions. What does this work say about (or how does it relate to) 'free' charged leptons?

A reference should be provided for the order of magnitude estimate of alpha* especially as this estimate is essential for the model verification in Eq. (35).

A reference should be provided for the ‘surface-tension-like’ force in Eq. (16) as it determines the existence of three coefficients and thus possibly three stable charged droplets (section 12).

Do not use types of electron, or ‘common’ electron versus others, but ‘charged leptons’

Replace ‘perturbation’ terms by ‘series expansion’ terms. The former has a different meaning in physics.

Additional suggestions:

Title: ‘solved’ sounds too strong, although in a question. Please consider something like ‘addressed’.

Make reference to charged lepton mass by coupling to Higgs field, consider e.g. EmQM conference references, and include references for the ‘matched asymptotic expansion method’ and for ‘radiation resistance’ (Larmor).

A better sectioning of the article, including subsections, would strongly increase readability. The author should consider removing sections 5 to 9 (possibly by adding some minor info in other sections), as they do not contribute to the conclusions, which are essentially on the equilibrium states of the pulsating droplet. Section 11 could be removed as well.

Asterisks seem to be missing in Eq. (21), and possibly in following equations. Please verify.

Arriving at Eq. (27) from (25) and (26) is not fully clear. Please expand on lines 404-405.

Line 441: there is no m defined in section 15, so please clarify where this parameter originates.

The introduction of beta in Eq. (34) is not necessary and confusing. Pi times sqrt(1/10) approximately equals 1, which is sufficient for the conclusions.

Author Response

Thank you very much for your careful review. It has resulted in many corrections and improvements of the paper. Details are given in the attachment.

Reviewer 2 Report

This paper considers, in the context of Emergent Quantum Mechanics, a droplet model for the electron. As mentioned by the author, this is reminiscent of the bouncing droplets model of Quantum Mechanics, in turn related to the Pilot-wave of de Broglie and Bohm. The main result is an ingenious model reproducing the masses of the electron, muon and tau particles.

Even if in a more advanced research, the analysis should consider the predictive power of such a theory with respect to the one of the Standard Model, I feel that the present investigation contains some preliminary interesting results. However, in my opinion, the first sentence in the abstract is a bit misleading, "...why electrons can have three different masses" seems an excessive poetic license, the electron has only one mass. As such I recommend the paper for publication Applied Sciences but the author should change the above sentence.

Author Response

Thank you very much for your time spent reading the paper and preparing your comments. I have removed  the "poetic freedoms in terminology" (a pity, but I agree that they are not appropriate in a scientific paper).

Reviewer 3 Report

Review Report

The Enigma of the Muon and Tau Solved by Emergent Quantum Mechanics?

My opinion of this work is that it is unsuitable for publication in this journal.  I will point out some of the significant items which I consider to be major flaws of this work.

(1)   The first sentence of the abstract is inaccurate, where the author refers to the long standing question of why electrons can have three different masses.  A more appropriate term is the word “leptons” in lieu of the word “electrons”.

(2)   After that, the whole rest of the paper attempts to build upon this hypothesis by applying classical physics and rudimentary electromagnetism in order to provide justification for different masses for electrons, when the mass of the electron has been firmly established through quantum theory and experiment.  The paper completely neglects the fundamental principles which could give rise to, what can more precisely be identified as three different families of leptons, namely the electroweak interaction.  None of the references or analyses take this into account, or even quantum mechanics and quantum field theory, which through gauge invariance, symmetries and things such as the Yukawa coupling could be studied as more plausible mechanisms for the existence of mass eigenstates for the leptons, which are themselves fundamental particles of nature.  The author is attempting to attribute the existence of mass solely to the electromagnetic interaction, and the only conserved quantum number for this interaction is the electric charge.  If one extracts the electric charge from the discussion, then the work fails to address the phenomenon of neutrino oscillations, which deals with the existence of mass unrelated to charge, and could be of relevance to the existence of these three families.  A more plausible explanation for the existence of masses cannot be found solely within the electromagnetic interaction.

(3)   Next, I will proceed to some of the technical aspects of this work.  For instance, the collection of references is purely insufficient to provide adequate background necessary to have an appreciation of the scope or the complexity of this subject.  Additionally, there are very few scientific papers from peer-reviewed journal provided, and mostly books which are more likely difficult for the reader to find due to their obscurity.  For any scientific work to be considered, modern, recent and relevant, easily accessible  documents should be an integral part of the work.  This is not the case here.  As another example, the author makes numerous references to Ref[7], which contains calculations “too lengthy” to be displayed in the work, and the results of these calculations are highly relied upon, with no guidance from the work itself as to the context or organization.  Any calculations this widely relied upon should be independently derived within the context used, so that the work is more self-contained.  This is all the more important when one is delving into a topic which is as highly speculative as this one, for which there is not a plethora of available background or documentation.

(4)   The work hinges on attempting to identify the lepton masses with equilibrium configurations associated to a droplet of electric charge.  Ignoring the fact that the electron is already known to be a fundamental particle with quantized electric charge, and not a composite particle such as a droplet with a continuous charge distribution, then the work is still highly speculative to a deeper level beyond the one aforementioned.  The model utilized has too many simplifying and unjustified assumptions, based upon classical physics which neglects the advances in quantum mechanics, and the analysis omits numerous important steps needed to verify its validity (within the confines of the inaccurate paradigm forming the basis of the paper).  An electron is not a charge distribution with rotational symmetry, with one-dimensional motion as the author uses.  Again, in section 4, the author relies heavily upon reference [7], in attempting to justify an electromagnetic mass.  A more accurate derivation of that concept is not Eq.(3), but rather an integration over all space of the energy density associated with the electric field of an electron, which should be included with the inertial mass. 

(5)   There are too many simplifying assumptions relied upon, even within the restriction of the faulty paradigm.  Equation (16), the expression for surface tension, is used with the dynamical equations (10) through (20) as a basis to come up with a cubic equation whose roots will denote the three electron masses.  No numerical analysis is provided of the positivity of the roots, although Fig.3 seems to be a useful graph for conceptualizing what the author is doing.  However, the expression in Eq.(16) appears to be just part of a function of s which should in principle have an infinite number of terms when carried out to all orders, which is nonanalytic at s=0.  Truncation of this function to three terms, corresponding to three masses, has no physical basis other than an attempt to make a link to the masses of the other leptons which is completely speculative- whose masses mind you are more plausibly (as previously indicated) attributable to interactions other than the electromagnetic interaction which is considered here.

(6)   Finally, the nebulous association of this work to quantum mechanics via “emergent” quantum mechanics appears to be simply an exercise in fine-tuning, and it is misleading to imply that one has arrived at Planck’s constant from fundamental classical arguments.

For these reasons, in my view the work is unsuitable for publication, and does not do justice to the highly complex factors required for such an analysis.  An attempt to introduce a new concept for which there is a dearth of background necessitates a proportionate respect for background information, deference to the existing body of scientific evidence and results, and a more comprehensive analysis of the relevant theory (in this case quantum field theory) and regimes of relevance.  This is not done here- and the scope of the work provided is inadequate to do deference to a topic of this magnitude- and even within the suspension of disbelief required in order to read through the article- lacks the necessary mathematical rigor.

Author Response

The usual procedure is: working hypothesis --> mathematical analysis --> conclusions compared with experimental data. In the case of the paper the experimental data are clear and agree with the conclusions. Any valid criticism can thus only concern the analysis of the droplet model. Nowhere in the review report my analysis (mind: within the context of Emergent QM) has been dealt with. The numerous, extensive expose's in the review report about the views of usual QM on the problem (quantum  field theory is mentioned in particular) are therefore, although interesting, beside the point that really matters. Errors in the analysis of the model have not been pointed out. See the attachment for more details.

Round 2

Reviewer 3 Report

While the author has somewhat addressed one, of several, of my concerns- as the author has acknowledged that we are indeed talking about leptons rather than just different types of electrons with different mass and has modified the recurring parts of the article accordingly, it only highlights more the reasons why the article should be rejected.

- The author would like to base the possibility of different masses for the leptons purely upon the electromagnetic interaction, using, essentially, a liquid drop model of charge.  Such an approach is not grounded in scientific reality:

(1) The electron is a fundamental particle of nature, and its electric charge is a fundamental quantum number, which has been measured and confirmed by experiment- this is well accepted.  The author is proposing something more akin to the liquid drop model sometimes used in nuclear physics, where one is describing a nucleus of composite particles which as well have a nontrivial magnetic and electric dipole moment- this is not what an electron is.

  So to start with, the model being considered ignores the standard developments for physics for the whole last century.

(2) The concept of why charged particles can have different masses actually has a firm foundation.  Heisenberg came up with the idea of isospin, wherein one can consider the proton and the neutron as different states of the same particle.  They have nearly equal masses and different electric charges.  If one were to switch off the electromagnetic interaction, then the aforementioned symmetry would be exact or nearly exact.  Thus the electromagnetic interaction prescribes the scale at which this symmetry becomes broken yielding the different particles since its Hamiltonian fails to commute at that scale with the generators of the symmetry.  The difference in masses could perhaps be attributed to a certain extent to electric charge in this case.  And the electric charges of the particles in the multiplet are the eigenvalues of the charge operator, which is a linear combination of isospin and hypercharge.

  In the isospin model the charges of the multiplet are different and the masses are different.  The author is attempting to study a model in which the charges, right off the bat, are the same- this already negates any possibility (based upon modern physics, field theory and quantum mechanics) that the differences in mass can be be correlated with electric charge- but electric charge is necessary for the mediation of the electromagnetic interaction being considered- which for example in SU(2) isospin is responsible for the differences in charge and to an extent in mass.  So, this rules out the thesis of mass differences as being due to electromagnetism. 

(3) The author is attempting to make inferences on particles described by quantum mechanics of weak interaction physics based purely on classical electromagnetism.  There is zero input from field theory and the symmetries that govern the interactions of the particles in question (as electromagnetism has been ruled out).

For these main reasons, the article is not suitable for publication- it is not a realistic or relevant model to the phenomenon being studied, and neglects the physics most likely to be relevant.  So the results cannot reflect the reality of the leptonic masses- since the physics of leptons is not taken into account.